# An Experimental Assessment of Depth Estimation in Transparent and Translucent Scenes for Intel RealSense D415, SR305 and L515

**DOI:** 10.3390/s22197378

**Published:** 2022-09-28

**Authors:** Eva Curto, Helder Araujo

**Affiliations:** Institute of Systems and Robotics, Department of Electrical and Computer Engineering, University of Coimbra, 3030-290 Coimbra, Portugal

**Keywords:** RGB-D cameras, range sensors, Intel^®^ Realsense™, LiDAR, structured-light, active stereoscopy, depth estimation, error sources, transparency, translucency, scattering, refraction

## Abstract

RGB-D cameras have become common in many research fields since these inexpensive devices provide dense 3D information from the observed scene. Over the past few years, the RealSense™ range from Intel^®^ has introduced new, cost-effective RGB-D sensors with different technologies, more sophisticated in both hardware and software. Models D415, SR305, and L515 are examples of successful cameras launched by Intel^®^ RealSense™ between 2018 and 2020. These three cameras are different since they have distinct operating principles. Then, their behavior concerning depth estimation while in the presence of many error sources will also be specific. For instance, semi-transparent and scattering media are expected error sources for an RGB-D sensor. The main new contribution of this paper is a full evaluation and comparison between the three Intel RealSense cameras in scenarios with transparency and translucency. We propose an experimental setup involving an aquarium and liquids. The evaluation, based on repeatability/precision and statistical distribution of the acquired depth, allows us to compare the three cameras and conclude that Intel RealSense D415 has overall the best behavior namely in what concerns the statistical variability (also known as precision or repeatability) and also in what concerns valid measurements.

## 1. Introduction

Since 2010, when Microsoft, in cooperation with PrimeSense, released the first Kinect, consumer RGB-D cameras have been through a democratization process, becoming very appealing to many areas of application, such as robotics [1,2], automotive [3], industrial [4], augmented reality (AR) [5], object detection [6], 3D reconstruction [7], and biomedical field [8]. All these applications thrived by receiving depth information in addition to color. To this fusion, between a typical color camera (RGB) and a depth sensor (D) in one unique device, we call an RGB-D camera—also known as range imaging camera or range sensing device [9].

Human–machine interaction applications were considerable drivers of the generalized application of RGB-D cameras, mainly of active technology which are the ones discussed in this article. In consequence, most of these devices were designed for the purpose of detecting and tracking bodies, or, more generally, to obtain the shape of opaque bodies/surfaces, through the analysis of the diffuse reflection components [10]. At the same time, there is vast interest in using these cost-effective cameras in other tasks related to robotics and 3D reconstruction, where the 3D perception of the targets can experience the most varied optic behaviors, from the basic, such as reflection and absorption, to the most complex, such as refraction, scattering and diffraction. Moreover, not all objects have a diffuse nature and so light does not come back to the sensor in an expected way. In the case of transparent or translucent objects, light not only reflects at the surface but also transmits into the object, and it causes multiple reflections and transmissions inside the object [11]. Therefore, 3D acquisition of transparent/translucent objects is a difficult problem to solve despite some work having already been performed to address this issue [12,13,14].

In this paper, we analyze three state-of-the-art Intel^®^ RealSense™ RGB-D cameras. More precisely, the models SR305, D415, and L515, which operate with active depth sensing. Although they all employ active technology, the operating principles and technology are different and specific to each camera. The SR305 uses Structured Light (SL) technology, D415 relies on Active Stereoscopy (AS), and L515 works with Time-of-Flight (ToF) principle.

To evaluate and compare the depth estimation performance of the three different cameras in challenging scenarios comprising transparency, semi-transparency, and refraction/scattering, we performed a thorough experimental assessment.

For that purpose, an acquisition scene involving an aquarium and liquids was considered. Firstly, previous work related to the comparison and evaluation of RGB-D cameras is covered. Secondly, we describe the three operating principles and the technology behind the three RealSense™ models. Next, our (experimental) methodology is explained, as well as all the experimental setup and the evaluation approach. The evaluation is based on repeatability/precision, statistical distribution of the acquired depth and fitting point clouds to planes. Results are shown and discussed in the last section.

The contributions of this paper include:(1)We explain/review the operating principles of three different state-of-the-art RealSense™ RGB-D cameras;(2)We present an experimental framework specifically designed to acquire depth in a scenario where a transparent aquarium fills the entire field of view and translucent liquids are added to it;(3)We assess the depth estimation performance based on the number of points per image where the cameras failed to estimate depth;(4)We evaluate the depth estimation precision in terms of point-to-plane orthogonal distances and statistical dispersion of the plane normals;(5)We qualitatively compare the distribution of the depth between the three cameras for the seven different experiments.

Essentially, we evaluate and compare the depth estimation performance of three state-of-the-art RGB-D cameras with different operating principles in scenarios with transparency and translucency.

## 2. Related Work

Modern RGB-D cameras are very appreciated for applications in robotics and computer vision due to their compactness and capability to gather streams of 3D data in real-time with a frame rate of 15–30 fps and resolutions up to 1280 × 720. They could serve as low-cost 3D scanners as long as their accuracy and precision meet the application requirements. Therefore, it is fundamental to test, analyze, and compare new devices with different state-of-the-art technologies to identify each sensors’ weakness, understand how a given operating principle influences depth estimation and accuracy, and infer which device is more suitable for a given task. The computer vision community has investigated many approaches to evaluate and compare devices’ performance. Thus, we present a brief overview of related work that assesses and compares the depth’s precision of different RGB-D devices and respective technologies, using metrological or other literature-based techniques.

Beginning with a study about one of the most influential RGB-D devices—the Microsoft Kinect I—Menna et al. [15] evaluate, in a geometric perspective, the accuracy in modeling indoor environments by measuring a special rigid test-field with a calibrated reference scale bar. According to the authors, this structured light-based device achieved satisfactory metric performances for close-range applications.

A comparison between Structured Light and Time of Flight cameras—the Kinect I and the SR4000 from MESA Imaging—was provided in Hansard’s book about ToF cameras [16]. In order to analyze systematic and non-systematic sources of error in ToF sensors, they propose a performance evaluation framework based on depth-maps acquisitions and their ground truth images. This experimental framework aims to investigate not only the accuracy of the depth but also the sensitivity of the two types of sensors to the object materials (to their reflectivity). Therefore, they categorize the objects according to the material type. According to the authors, the most challenging category is the one for translucent objects, concluding that the RMSE drastically increases with the increment of translucency.

Following the release of the second version of Kinect, whose operating principle is Time of Flight, several studies evaluating and comparing both devices and, therefore, their technologies were conducted. In [17], Kinect II and its predecessor are compared through a metrological approach, more specifically, using previously calibrated standard artifacts based on 5 spheres and 7 cubes. They also tested Kinects’ accuracy and precision, varying ranges and inclination between each sensor and the artifact. Breuer et al. [18] provide a detailed analysis of Kinect II regarding measurement noise, accuracy, and precision. Their experiments show a significant distance dependent error that can be mitigated with further calibration. Furthermore, they mention that Kinect II can better resolve fine structures even though its resolution is lower than Kinect I. The two versions of Kinect are also compared and analyzed by Zennaro et al. [19]. They first evaluate the accuracy of the two sensors on near and far range test sets in an indoor environment, using different scenes and light conditions. Then, in outdoor tests, they analyzed the sensitivity of each sensor to the sunlight, concluding that, due to the different operating principles, Kinect II can better tolerate sunlight. Comparing the effectiveness of both sensors in two different applications (3D reconstruction and People Tracking) it was verified that the accuracy in both applications is better for Kinect II. In order to understand the suitability and accuracy of RGB-D cameras for close range 3D modeling, Lachat et al. [20] use Kinect II to reconstruct small objects in three dimensions. They use a reference point cloud acquired by the FARO ScanArm (ground truth) to qualitatively compare with the point cloud resulting from the Kinect II. A detailed comparison between Structured Light Kinect and Time of Flight Kinect is presented by Sarbolandi et al. [21]. The authors propose a framework to evaluate the two types of range sensors based on seven experimental setups covering all the main error sources for SL and/or ToF cameras. The effects of translucent material on the sensor’s performance were also analyzed in one of their test scenarios inspired by the experimental setup of [16]. Based on the results obtained, they outline a comparison of device performances in different conditions regarding the main source of errors to provide users hints for device selection. To explore the applicability of low-cost RGB-D cameras in standard 3D capturing problems, Guidi et al. [22] analyze systematic and random errors for five different devices featuring three different technologies. A systematic comparison of the Kinect I and Kinect II is given by [23]. The authors investigate the accuracy and precision of the devices for their usage in the context of 3D reconstruction, SLAM, or visual odometry. They take into account, for each device, factors such as temperature, camera distance, and color. Errors such as flying pixels or multipath are also investigated. They based the analysis of the accuracy and precision of the devices using as ground truth measurements of a planar wall, detected through the pose estimation of a checkerboard on the wall.

Driven by some academic prototypes building low-cost depth cameras based on coded light technology [24,25,26], Intel^®^ RealSense™ launched the world smallest consumer depth camera, the F200. Later, Intel^®^ proposed an improved version, the SR300 model [27]. Its technology, hardware, and algorithms were described in detail in [28]. This device, designed for short range, is composed of an IR pattern projector and an IR camera that operate in tandem using coded light patterns to produce a 2D array of monochromatic pixel values. The embedded imaging Application-Specific Integrated Circuit (ASIC) processes all these values to generate depth or infrared video frames. Being interested in the potentiality of this sensor as 3D scanner, Carfagni et al. [29] evaluated its performance using metrological and non-metrological techniques. VDI/VDE norms are applied to a set of raw captured depths and to a set of depths acquired with an optimized camera setting. In addition, the “real life” conditions for the use as a 3D scanner are reproduced using an optimized sensor setting. The performance of the device is critically compared against the performance of other short-range sensors. The survey of Giancola et al. [30] gives us a comprehensive metrological comparison of Time-of-Flight, Structured Light, and Active Stereoscopy Technologies. The cameras chosen to be characterized were: Kinect II (ToF), Orbbec Astra S (SL), and D415 from Intel^®^ D400 family. Their evaluation was focused on pixel-wise characterization and sensor-wise characterization. In order to obtain the ground truth distances for the pixel-wise characterization, two sophisticated setups with anthropomorphic robots were used: the first one, used with ToF and SL sensors, comprises the camera positioned on a tripod and aligned with the planar white target mounted on the anthropomorphic robot end-effector; the second, used with D415 camera, requires two robots that coordinate their motions between each other. One of the robots fixates the camera on the end-effector and the other moving the target. In addition to these experiments, they analyze the performance of the depth sensors in what concerns the measurement of objects with simple geometric shapes (plane, cylinder, sphere), the quality of the edge reconstruction and the influence of the incidence angle on the target. For the case of the ToF camera, the impact of the target color and material was also tested in what concerns the depth measurements and on the multiple path problem. Inspired by [29], the same work group [31] investigated the potential of the newly released Intel^®^ RealSense™ D415 as a 3D scanner. They carried tests that integrate VDI/VDE norms with literature-based strategies and provide a characterization and metrological evaluation for D415 camera. These results are also compared with the performances of other close-range sensors in the market. In the book *RGB-D Image Analysis and Processing* [32], Rodríguez and Guidi [33] dedicate one chapter to data quality assessment and improvement for RGB-D cameras, analyzing how random and systematic 3D measurement errors affect the global 3D data quality in the various operating principles. An experimental evaluation of three state-of-the-art Intel^®^ RealSense™ cameras was performed by Lourenço and Araujo [34]. The cameras SR305, D415, and L515 were tested using depth images of 3D planes at several distances. Their measure of accuracy is based on point-to-plane distance and the precision based on the repeatability of 3D model reconstruction. Recently, Breitbarth et al. [35] presented a practical assessment of the LiDAR Intel^®^ RealSense™ L515. They followed the VDI/VDE Guideline 2634 to perform the metrological characterization. According to this guideline, they define three quality parameters: probing error, sphere-spacing error, and the flatness measurement error. They conduct additional experiments on different measuring distances for plane objects with different surfaces. Moreover, results regarding the measurements of angles and the influence of the surface characteristics are also presented.

In [36], the most advanced international standard regulation—ISO 10360-13:2121 is used to evaluate and compare the performance of three RGB-D cameras from a metrological perspective. Furthermore, systematic errors were analyzed for the case of the acquisition of images of flat objects and also for the case of 3D reconstruction of objects. The sensors chosen for this study were the Intel^®^ RealSense™ models D415, D455, and L515. Although the authors tested three cameras, in practice, only two different operating principles were tested, as both the D415 and D455 work by active stereoscopy.

Most comparison studies between RGB-D cameras are focused on the metrological characterization of sensors. All these analyses are based on the assumption that the sensor is in a transparent medium (air) observing a non-transparent target. That is, it is assumed that light rays reflected by the scene objects travel to the observer without being affected by reflections and/or without crossing different physical media. However, since RGB-D sensors are used in various applications, observed scenes can contain transparent, reflective, and refractive elements that affect depth estimation.

This paper aims to compare the depth estimation performance of three RGB-D devices with different operating principles in scenes containing transparency. We propose an experimental assessment involving scenarios with well-defined transparency levels.

## 3. Fundamentals

### 3.1. Operating Principles and RealSense Cameras

Optical depth sensing techniques can be divided into passive and active approaches [30]. Passive methods rely on the reflection of radiation (typically in the visible spectrum) naturally present in the scene to infer the 3D distance measurement [37]. Thus, passive systems do not interfere with the scene to obtain its depth. A well-known example of this class is the stereo-vision system, which is inspired by the human binocular vision system. This approach uses as an input the 2D images of two (or more for the generalized case: N-view stereo system) monochrome or color cameras located slightly apart from each other. Depth information can be obtained by establishing correspondences between pixels from different cameras and then using triangulation [9]. The main drawback of passive systems is the high dependence of texture in the scenes. This issue does not apply to active vision systems since they use an external lighting source to provide depth information [30]. The three RGB-D cameras addressed in this work use active approaches, based on the use of IR projectors. The core of this work is to evaluate and compare the depth estimation performance of the three cameras in transparent and translucent scenarios. Therefore, it is pertinent to provide an overview of the technology behind each camera. In Figure 1, we can see the three models considered in this paper. On the left we have the D415 model which uses active stereoscopy technology; in the middle the LiDAR camera L515 whose operating principle is based on indirect time of flight; and in the right the SR305 model, a structured light depth camera.

#### 3.1.1. Structured Light

As aforementioned, passive methods might fail in featureless zones, since the correspondences needed for the triangulation process may not be enough. Structured Light (SL) technology addresses this issue by using light patterns to facilitate the correspondence problem. Differently from the stereo-vision system, SL substitutes a camera with a laser projector. Figure 2 illustrates the projector emitting various coded patterns, which, in turn, introduce artificial features into the scene being captured by the camera. The pattern reflected from the scene back to the camera will distort according to the shape variation of the scene. That is, for planar targets, the pattern remains the same [38]. By decoding the pattern, the correspondences between the projected pattern and the pattern captured by the camera can be established. Since we have correspondences, their displacement gives us a disparity map, which, similarly to stereo-vision, is used to estimate depth [39]. Considering that the focal length *f* of the camera and the baseline *b* between the projector and the camera are known, the depth of a given pixel (x,y) can be estimated using the respective disparity value m(x,y) for this pixel as
(1)$$D=\frac{b.f}{m(x,y)p},$$
being *p* the pixel size in length units [21].

The design of patterns is a crucial piece in the Structured Light systems. Therefore, several techniques have been proposed to guarantee high-resolution and high-accuracy in 3D acquisition [40]. Time-coding and Space-coding are the two fundamental categories: in the first, multiple patterns are sequentially projected to identify each point in the scene with a times series; in the latter, a single pattern is emitted. Time-coding should be used if high accuracy is a requisite. For dynamic scenes, spatial coding is more suitable, although it implies a lower spatial resolution [41].

Regarding the pros and cons of the SL technology: the non-dependency of the scene’s features and the capacity to achieve high resolution and high accuracy in 3D reconstruction turn Structured Light methods very appealing. On the other hand, external factors, such as ambient illumination and the reflectance and transparency of the surfaces drastically affect the performance of Structured Light sensors [42]. Additionally, its triangulated-based principle implies that the performance is affected by the physical dimensions and by the range. Strictly speaking, the measurement uncertainty is directly related to the size of the baseline and increases with the square of the range.

##### Intel^®^ RealSense™ SR305

The Intel^®^ RealSense™ SR305 contains the same depth module board as the SR300 camera since it is basically the SR300 redesigned as an entry level depth camera, more affordable and suitable for applications in robotics than the SR300 camera. Table 1 presents the main specifications of SR305. Of the three cameras, it is the one with the lowest maximum depth resolution (640 × 480) and the shortest measurement range (0.2–1.5 m). Being an SL camera, SR305 uses multiple different coded light patterns that change over time [43]. Specifically, a time-multiplexing code, called Gray Code (GC), which is a particular type of binary code [28], is used. Gray code patterns are a set of binary patterns coded by intensity, e.g., dark for 0 and bright for 1. Every adjacent code word with $2^{N}$ bits differs by only one bit, where *N* is the sequence number of the patterns. In this way, GC minimizes errors allowing us to obtain accurate correspondences between camera pixels and projector. In Figure 3, an example of Gray code patterns for N=3 [44] is shown [45].

The SR305’s IR projector comprises an 860 nm laser diode emitting a small spot laser beam, a lens that expands the beam into a vertical bar, and a MEMS mirror that horizontally scans the patterns. Therefore, while the scene is illuminated with sequential patterns, the IR camera captures the 2D array of monochromatic pixel values that will be then processed by the vision processor to generate a depth frame.

An exemplar of a colorized depth frame obtained by the SR305 camera is shown in Figure 2.

#### 3.1.2. Active Stereoscopy

Active stereoscopy, like passive stereoscopy, comprises two ordinary cameras that see the same scene and match the pixels between both to estimate depth by triangulation. The differential in AS lies in the features added to the scene through an unstructured light pattern. These artificial features are significant for low-texture surfaces, allowing a better reconstruction once we have many more correspondence points to triangulate [30,33]. The triangulation principle applies in a similar manner as in SL, so the Equation (Equation 1) also applies to AS. Consequently, we have to consider the limitations related to the triangulation-based principle (baseline and range affecting the measurement uncertainty). The two cameras make AS systems less sensitive to non-idealities in sensors and projectors and also to ambient light, since the two patterns being compared will be affected by the same factors in equal proportion [42]. In Figure 4, an illustration of the active stereoscopy technology is shown.

##### Intel^®^ RealSense™ D415

The D415 model is part of D400 series which offers complete depth cameras integrating vision processor, stereo depth module, RGB sensor with color image signal processing, and Inertial Measurement Unit (IMU) in the case of D435 and D455 models.

The active stereoscopy technology of D415 consists of left and right imagers and two AMS Heptagon projectors placed side by side used to improve texture-less scenes by projecting a non-visible static IR pattern with wavelength of 850 nm [46]. In Table 1, we give some of the specifications of D415 model, of which we can highlight the high-resolution depth (up to 1280 × 720 pixels at 30 frames per second), the long-range capability and a field of view well suited for high-accuracy applications [31,47].

Examples of IR cameras frames of the D415 stereo system can be seen in Figure 4, as well as a resulting depth frame, colorized in the function of depth values.

#### 3.1.3. Time of Flight

The Time of Flight technology, as the name suggests, is based on measuring the time it takes from emitting a signal until the reflection is received. Since the speed of light *c* is a known constant, the distance between camera and object can be estimated [16,30,32,42]. There are two different types of ToF technology (depending on the measuring mode) which are categorized into direct Time of Flight (d-ToF), also called Pulsed ToF, and indirect Time of Flight (i-ToF), also known as Modulated ToF.

Direct ToF technique is based on transmitting a light pulse, receiving the reflection and then measure the delay between the emitted and the received light pulse, Δt. The scene depth can be estimated as half of the measured round trip distance:(2)$$D=\frac{c\times \Delta t}{2}.$$

On the other hand, in i-ToF, instead of measuring the delay of a single pulse trip, a continuous modulated light is transmitted and the phase difference between outgoing and incoming signals is measured. The phase shift (ΔΦ) estimation is completed in a robust and precise way using cross-correlation techniques. The distance can be calculated according to:(3)$$D=\frac{c\times \Delta \Phi }{4\pi f},$$
where *f* is the modulation frequency. In this paper, we analyze the i-ToF principle present in L515 camera from Intel^®^ Realsense™.

Contrarily to structured light and active stereoscopy technologies, ToF systems have the advantage of not being affected by their physical dimensions nor by significant occlusion. The reason for the latter is that the transmitter and receiver are designed such that they are collinear, that is, the transmitter is placed on a line close to the receiver [42]. A drawback of this system is the dependency on the reflectivity of the scene. ToF is not ideal for dark surfaces (black/gray/shiny). Additionally, challenging lighting conditions (e.g., outdoor environment) can limit the range. One common issue with ToF sensors is the multiple propagation paths from the light source to pixel, commonly called multi-path interference. This phenomenon can result from inter-reflections in the scene and/or the lens camera system, subsurface scattering, and mixed pixels due to the solid angle viewed by a pixel covering two distinct objects [48]. Of the three technologies presented here, this is the most expensive.

##### Intel^®^ RealSense™ L515

Intel^®^ RealSense™ presents its only LiDAR camera as a highly accurate depth sensing able to fit in a palm of a hand. This camera relies on the i-ToF technology explained above. More precisely, while a pulse is continuously transmitted and received, multiple frequencies are cross-correlated with a known template. This correlation generates recognizable peaks that represent the best match and the best accuracy for a specific point and this is also the value that is then transmitted.

The L515 device comprises a transmitter, a receiver and a depth module. The transmitter consists of an Edge-Emitter Laser (EEL) that modulates at 500 Mbps (megabits per second) a continuous IR code (at 860 nm) and an advanced 2D MEMs (micro-electromechanical system) mirror that moves the laser across the field of view [49,50]. In the receiver the return light is collected by a single avalanche photodiode (APD) featuring low input referred noise (IRN). This omni receiver is able to collect the light of the entire scene so it does not follow the laser beam but it collects photo photons coming in from the full scene. The depth module, a custom-made ASIC, is where the visual processing takes place, comprising all cross-correlation algorithms, depth calculations, and post-processing filters. Figure 5 schematizes the ToF technology present in the L515 camera.

Since L515 measures the depth at each pixel in a direct way, its edge resolution is higher compared to that of other depth sensors, such as stereo cameras. The L515 outstands from conventional ToF sensors, featuring a reduced multi-path effect. For instance, in a scenario where we have the intersection of two walls (a corner), the L515 shows straight walls with a small amount of curvature at the corner, while a conventional TOF camera shows bowed walls and a much larger curvature at the intersection [49]. On the other hand, L515 is highly dependent on the signal to noise ratio (SNR), meaning that a change in the quality of the returned signal significantly affects the depth results.

The principal specifications of L515 can be found in Table 1.

### 3.2. Light Interaction with Transparent and Translucent Objects

As described above, active approaches depend (each in their way) on the returned light then captured by the camera. The ideal conditions for which 3D active systems are designed are the air as propagation medium (of the active light) and considering Lambertian targets [51,52]. Their performance may be affected in non-ideal conditions by the nature of the materials involved, or more precisely, the way that light interacts with them [45]. When a light wave encounters an object, we can expect some phenomena, such as transmission, reflection, absorption, refraction, scattering, and others, depending on the composition of the object and the wavelength of the light. We can define materials in terms of reflectance, absorbance, and transmittance. Transmittance of a material, that is, its effectiveness in transmitting light, is the least studied property compared to the others [53].

For the sake of clarity (in this paper), there are three concepts of optics and human perception that must be clarified: transparency, opacity, and translucency. Roughly speaking, an object is described by human perception as transparent if we can see things behind that object (high transmittance) and as opaque otherwise (low transmittance) [54]. Despite transparency and translucency being occasionally used interchangeably, they mean different concepts. Translucency is understood as the intermediate phenomenon between transparency and opacity [55].

In geometrical optics terms, a transparent surface produces both reflected and transmitted light rays, as illustrated in Figure 6. Transparent objects transmit light without diffusing it, whereas translucent ones transmit light diffused in all directions which results in a “blurred” image of the background [55].

Translucent objects exhibit both surface and subsurface scattering (or volume scattering). Moreover, light is propagated from one point of incidence through the object’s interior, emerging at different positions. For these reasons, the optical properties of translucent objects are complex to analyze. The translucency of a material depends on intrinsic and extrinsic factors. The intrinsic factors define how the light propagates thought the media. They are the refractive index, the absorption and scattering coefficients, and the scattering phase function. The extrinsic factors can be, among others, the illumination direction, the object shape/size, the surface roughness, and the color of the background behind the object. For instance, large translucent objects will transmit less light than small translucent objects made of the same material and shape [53].

When a light beam is incident upon a transparent surface, part of it is reflected and part is transmitted through the material as refracted light, as illustrated in Figure 7. Because the speed of light is different in different materials, the path of the refracted light is different from that of the incident light. The direction of the refracted light, specified by the angle of refraction with respect to the surface normal vector, is a function of the index of refraction of the material and the incoming direction of the incident light. The angle θ of refraction is calculated from Snell’s law as
(4)$$sin\theta_{r}=\frac{\eta_{i}}{\eta_{r}}sin\theta_{i}$$
where θ is the angle of incidence, $\eta_{i}$ is the index of refraction for the incident material, and $\eta_{r}$ is the index of refraction for the refracting material. The overall effect of the refraction is to shift the incident light to a parallel path as it emerges from the material (when the boundary surfaces are parallel).

As mentioned, translucent materials exhibit scattering. The main observable consequences of multiple scattering in volumetric media are spatial and angular spreading of the incident light distribution, as could be seen in Figure 8. The absorption effect is also present in many materials including water, however, the absorption is often orders of magnitude smaller than the scattering.

## 4. Materials and Methods

### 4.1. Experimental Setup

As stated in Introduction, the core of this work lies in comparing the performance of three different operating principles in depth estimation for transparent and refractive targets. For this purpose, Intel^®^ RealSense™ cameras were chosen, specifically the models SR305 (SL) [27], D415 (AS) [46], and the newest L515 (LiDAR) [49], all operating with the Intel^®^ RealSense™ SDK 2. The RealSense library librealsense2 served as a basis for all the depth acquisition software, that was coded in C++ and run in Ubuntu operating system. For each different test, 100 depth frames were obtained and saved. In addition to depth, data from the RGB and infrared cameras were also recorded. To ensure a fair comparison, we set the cameras with the same depth resolution 640 × 480, since it is the unique resolution of the SR305 camera. The remaining streaming settings were used in the default mode, except for the L515 camera, where the “Short Range” preset was selected. The acquisitions were made in a darkroom containing an adjustable LED board. In this way, all the acquisitions were performed under constant illumination. Our setup for the experiments with the aquarium was the following:A glass aquarium with dimensions 0.84 × 0.22 × 0.58 m (and about 6 mm of thickness);A table to support and elevate the aquarium so that the floor does not appear in the visual field;A flat white wall behind the aquarium that is partially covered with a black cardboard (Figure 9);A RealSense™ depth camera mounted on a tripod. The device is pointed to the aquarium, having the optical axes approximately perpendicular to the wall.

Figure 10 illustrates the configuration of the experimental setup and the distances between the camera, the aquarium, and the wall. The aquarium setup can be visualized in the pictures of Figure 11.

Our experimental framework includes the following tests outlined below:*Wall*: This is a zero test where we acquire depth images directly from the wall, that is, without any transparent objects between the camera and the wall. These depth measurements will serve as a reference to the following tests since we do not have ground truth for the depth.*Empty*: In this test, the aquarium is inserted between the camera and the wall. Therefore, we aim to analyze the influence of the two transparent glass walls of the aquarium (with air in-between) in the depth estimation.*Water full*: This test introduces another challenging scenario regarding transparency. The aquarium is filled with water (about 95 L). Then, between the camera and the wall, we have a first glass wall, water, a second glass wall, and air.*Water milk1/2/3/4*: Set of tests, where the water is dyed with milk to experience different levels of transmittance (transparency, translucency and opacity). *Water milk1* is the less opaque of the four, with a milk concentration of 0.03% (*v*/*v*). Then, *water milk2* with 0.13% (*v*/*v*) and *water milk3* with 0.24% (*v*/*v*) Finally, the most opaque solution, *water milk4* with 1.13% (*v*/*v*).

For each of these experiments, the cameras were sequentially switched. In Figure 11, we can see the pictures regarding the test *water full* and also the tests *water milk1/2/3/4*. Examples of depth frames acquired by the three cameras for each transparency scenario are shown in Table A1.

### 4.2. Experimental Evaluation

Since the camera SDK provides its intrinsic parameters, it was possible to convert the depth frames into point clouds [47]. Given the pixel coordinates and the respective depth measurements $d_{m}$ in an image with no distortion, the corresponding 3D points were computed. The *z* coordinates correspond to the depth values measured at that points, i.e., $z=d_{m}$. The coordinates *x* and *y* were computed as follows:(5)$$x=z\times \frac{u-pp_{x}}{f_{x}}$$
(6)$$y=z\times \frac{v-pp_{y}}{f_{y}}$$
where (u,v) are the pixel coordinates, $\left(pp_{x},pp_{y}\right)$ are the coordinates of the principal point and $f_{x}$ and $f_{y}$ are the focal lengths in pixel units.

An example of a depth frame converted to a point cloud is shown in Figure 12. Since a D415 acquisition is given as an example, it is visible in the frame’s leftmost area the invalid depth band, a D415’s particularity that is explained in [46].

#### 4.2.1. Depth Estimation Failure Rate

One way to evaluate the sensors performance in first instance is through the number of points per image where the cameras failed to estimate depth. These points correspond to the invalid pixels in the frames, i.e., pixels whose cameras cannot estimate a depth value and their value is set to zero by the Intel RealSense SDK. Considering that in this article the analysis is performed in relation to 3D points, instead of invalid pixels, we use the term *invalid points*. Therefore, the evaluation of the *invalid points* was conducted as follows:For each experiment (Camera-test), two Regions of Interest (RoI) were segmented—the *right band* (rb) and the *left band* (lb). The decision of having two different regions aimed at allowing the evaluation of the depth estimation as a function of the material reflectivity. Thus, the *left band* is a RoI with a black background (a cardboard) and the *right band* is a RoI with a white background (the wall). These areas were carefully selected (avoiding artifacts and the invalid band of D415) and defined in a frame (as illustrated in Figure 13a). A single frame out of 100 was used to select and define the two RoIs. The RoIs so defined were the same for all the frames of the same acquisition. Figure 13b shows the respective representation of the segmented regions in the generated point cloud.For each one of the 100 samples, the points in the RoIs whose depth estimation failed were counted and the percentage of *invalid points* was obtained. Then, the average of the 100 percentages for each experiment was estimated.

By using regions to do the analysis, we can evaluate and compare the performance of the cameras against the two different backgrounds on the wall. Due to the dependence of L515 and SR305 on the reflection of the IR pattern, they perform worse on dark surfaces, as expected. So, it would be unfair to evaluate everything homogeneously. Furthermore, this approach also leaves out the invalid band of the D415. In Appendix A, the segmented bands for all the Camera-test experiments are shown (Table A1).

#### 4.2.2. Depth Precision

The precision was evaluated in terms of point-to-plane orthogonal distances. Given the 3D structure of the scene, it was considered that a reference plane corresponding to the wall should be considered. Even though in some of the experiments points belonging to other planes were detected, and for each experiment, a single dominant plane was considered.

Therefore, the point clouds were fit onto a single dominant plane, i.e., the following plane equation was used:(7)$$xn_{x}+yn_{y}+zn_{z}-d=0$$
where $\left(n_{x},n_{y},n_{z}\right)$ are the components that constitute the unitary normal, (x,y,z) are the points coordinates and *d* stands for distance from plane to the origin.

Total Least Squares (TLSs) were used to estimate the plane parameters. The processing steps applied to the data are described below and illustrated in Figure 14 (for a single case). Additionally, Table 2 shows the structures of the point clouds.

***Coherent points*****filtering:** In this process, only the *coherent points* of each point cloud are saved to further be concatenated. We designate as a *coherent point* a point that in all the 100 point clouds meets the following condition 0m<z<0.7m, with *z* being the depth. That is, the *coherent point* depth has to be greater than zero, since the SDK sets depth to zero in case of an invalid depth measure. Additionally, the *coherent point* depth must be less than 0.7 m, given that the wall is 0.63 m away from the cameras and so points with depth beyond this value are physically impossible. The *coherent point* depth must be within this range for all 100 point clouds. Otherwise, this point is excluded and will not be saved in the concatenated point cloud. In Figure 14, we can visualize one of 100 point clouds in raw Figure 14a, named ptCloud_raw. Furthermore, the same point cloud after filtering for *coherent points* is in ptCloud_filt.**Concatenation:** We concatenate all the 100 point clouds into a single one. In other words, considering a filtered point $\mathrm{ptCloud}\_{\mathrm{filt}}_{i}$ with i=1,…,N and N=100, a filtered point cloud has a shape, such as $\left(3,n\left(\mathrm{ptCloud}\_{\mathrm{filt}}_{i}\right)\right)$, where $n(\mathrm{ptCloud}\_{\mathrm{filt}}_{i})$ is the number of points of the filtered point cloud *i*. The concatenated point cloud—the ptCloud_concat has the shape $\mathrm{ptCloud}\_\mathrm{concat}(3,\sum_{i=1}^{N}n\left(\mathrm{ptCloud}\_{\mathrm{filt}}_{i}\right)$. A point cloud ptCloud_concat resulting from the concatenation is shown in Figure 14c.**Removal of outliers:** Despite the previous *coherent points* filtering, there are still artifacts in the point clouds. In Figure 14c, we can observe a cluster of 3D points distant (in the Z axis) from a potential cluster to a plane. Therefore, it is important to remove outliers from the ptCloud_concat with respect to the depth values (the *z* coordinate of the 3D points). So, a point was considered an outlier if its *Z* coordinate is greater than three scaled median absolute deviations (MAD). The scaled MAD is defined as follows:
(8)$$C\times (median\left(\left|x_{i}-median\right|\right)),\;i=1,2,3,\ldots ,n$$
(9)$$C=\frac{1}{\sqrt{2ICE(3/2)}}$$
where $x_{i}$ is the depth value of the *i* point, *n* is the number of points and ICE is the Inverse Complementary Error [57]. Figure 14d shows the resulting point cloud ptCloud_in with inlier points.**Data centering:** Each set of points was centered in the X and Y axes so that their center of mass is positioned at the center of the XY’s plane of the camera’s coordinate system. Figure 14e presents the centered point cloud—ptCloud_cent.**Fitting point clouds to planes:** After the described processing steps, we have 21 point clouds (of type ptCloud_cent) corresponding to the three cameras and the seven different tests. Planes were then estimated for these point clouds. The Total Least Squares (TLS) method, also known as Orthogonal Regression method, was used to estimate the parameters. TLS minimizes the sum of the squared orthogonal distances between the points and the estimated plane [58]. In [59], we can find the fundamentals and the mathematical equations used to estimate the plane. For the practical estimation with the TLS method, the Principal Component Analysis (PCA) was used, based on the *MATLAB* example [60]. In Figure 14f, we can see an example of an estimated plane, as well as the orthogonal distances between the plane and the data points. In Appendix B, plots with the orthogonal distances for all cameras and tests are shown.**Analysis of the orthogonal distances:** The orthogonal distances are analyzed in terms of descriptive statistics (statistical measures of central tendency and statistical dispersion). The formulas used for the descriptive statistics of a set of observations $x_{1},\ldots ,x_{n}$ (which correspond to the estimated orthogonal distances) are the following:
Arithmetic mean
(10)$$\bar{x}=\frac{1}{n}\left(\sum_{i=1}^{n}x_{i}\right).$$Median
(11)$$\tilde{x}=\left\{\begin{array}{cc} \mathrm{the}\;\frac{n+1}{2}\;-\mathrm{th}\;\mathrm{element}, & \mathrm{if}\;n\;\mathrm{is}\;\mathrm{odd}\\ \mathrm{the}\;\mathrm{mean}\;\mathrm{of}\;\mathrm{the}\;\frac{n}{2}-\mathrm{th}\;\mathrm{and}\;\mathrm{the}\;\left(\frac{n}{2}+1\right)\;-\mathrm{th}\;\mathrm{element}, & \mathrm{if}\;n\;\mathrm{is}\;\mathrm{even}. \end{array}\right.$$Standard Deviation
(12)$$\sigma =\sqrt{\frac{1}{n-1}\sum_{i=1}^{n}{\left|x_{i}-\bar{x}\right|}^{2}}.$$Mean Squared Error (MSE)
(13)$$MSE=\frac{\sum_{i=1}^{n}{\left(x_{i}-{\hat{x}}_{i}\right)}^{2}}{n}.$$
where *n* is the number of the observations (in this case, the number of the orthogonal distances) and ${\hat{x}}_{i}$ is the corresponding predicted value by the model.

#### 4.2.3. Precision in Terms of Plane Modeling Consistency

In the previous sections, an estimated plane model was obtained for every camera-test experiment with the corresponding analysis being focused on the orthogonal distances. In order to assess the consistency of the plane parameters (especially the normal), we also carry out an additional analysis where instead of estimating the plane for the concatenated point cloud, planes are calculated for each and every one of the 100 point clouds (after filtering and with outliers removal). Once again, descriptive statistics (statistical measures of central tendency and statistical dispersion) were used to characterize the observations, that are, in this case, the estimated parameters. Moreover, directional statistics are employed on the planes’ normal vectors to better describe the model’s variation. All the formulas regarding circular statistics and spherical statistic are given in the Appendix C.

#### 4.2.4. Qualitative Analysis of Depth Distribution

In order to qualitatively evaluate the distribution of depth as a function of different materials with different degrees of transparency/translucency (glass, water, and water and milk solution), all the 100 acquisitions from each test were used. Having the point clouds, we estimated the average depth (i.e., the z coordinate) for each point (in the image) from the 100 samples. To exclude non-stable depth values, we did not estimate depths for those points where less than 80 positive depths (out of 100 measurements) were obtained. This data analysis allows us to qualitatively analyze the estimation of depth, as a function of the material.

## 5. Results and Discussion

The results presented in this section are organized according to the methodologies (described in the previous section) used to asses the depth estimation performance of the cameras in the configurations considered. The results are discussed within a framework of a comparison between the cameras performances.

### 5.1. Results and Discussion Regarding Depth Estimation Failure Rate

Table 3 shows the results for the average percentage of *invalid points* regarding the points existing in the *right band* and in the *left band*. The results show that the D415 camera outperforms L515 and SR305 since it has the smallest failure rates for each test and in both bands. As expected, the results show that the color background (being white or black) affects the failure rate of depth estimation, particularly for the L515 and SR305 cameras that are highly dependent on the laser reflection. As expected, the experiments with smaller failure rates, independently of the cameras, are the those corresponding to the *wall* test. The results for *empty* test and for the *water full* test have failure rates slightly higher. Regarding the tests with milk, the results are more complex to analyze. A higher concentration of milk in water leads to higher light scattering (Figure 8) and depth can not be estimated at bigger distances. In those cases (and due to the decrease in transparency) the first wall of the aquarium is detected and most depth estimates correspond to this first wall of the aquarium. In what concerns tests *water milk3* and *water milk4*, D415 presents low percentages of invalids points since the milk present in water is enough for the camera to detect the first wall of the aquarium. For the L515 camera, the average percentage in the *water full* test is almost 100% in the *left band*. That means that the L515 camera can not estimate depth for the full aquarium (with only water) when the background is dark. Adding milk to water, we observe a similar pattern in results as for D415. The SR305 exhibits the worst results. This camera is only capable of estimate depth in the *left band* for the *water milk4*. This is explained by the fact of, in *water milk4*, the opacity of the water–milk solution prevents the detection of the black cardboard visible (for the infrared patterns).

### 5.2. Results and Discussion Regarding Depth Precision—Orthogonal Distances

The statistical analysis of the orthogonal distances, obtained by estimating planes in the point clouds, is given in Table 4 and Table 5. In addition to the statistical parameters, the estimated distance from the camera to the plane and the percentage of points used for the plane estimation are also included in the table. The distance to the plane is relevant to determine which was the plane of the scene that was estimated (since the scene contains multiple planes namely the wall and the aquarium front and back walls). On the other hand, the percentage of points used is also pertinent since a greater number of points is advantageous to the quality of the estimates. The percentage of points used was calculated based on the initial number of points (from the initial point cloud without filtering) and the number of final points (after the processing steps described in Section 4.2.2). Due to the low percentage of points used in some of the acquisitions, the corresponding analysis was not performed.

In the first table (Table 4), we have the results for the experiments *wall*, *empty* and *water full*.

In the case of the *wall* test, the estimated distance from the cameras to the wall plane (one of the parameters computed by the TLS method) is approximately 0.63 m, which coincides with the measured distance between the cameras and the wall. The L515 camera has the biggest percentage of points used in the estimation (98%) whereas the D415 has less 9% (89%) and the SR305 has only 49%. This result from SR305 is due to the black cardboard existing in the background that absorbs the projected patterns. Regarding the statistical estimates, the smallest median of the signed distances median corresponds to D415 with an order of magnitude of O−5. For the median of the absolute distances, the three cameras have the same order of magnitude values (O−3), with the D415 having the smallest value. In what concerns the average of signed distance values, D415 has the biggest orthogonal distances average (which is very small O−19).

In the case of the average of absolute distances, the smallest average is for the D415 (O−3). Both the standard deviation (std) for signed distances and the standard deviation for absolute distances have the same order of magnitude (O−3) for the three cameras, with the D415 having the smallest value. In what concerns the estimated MSE (mean squared error) the smallest value is for the D415 with a value with an order of magnitude of O−6. Therefore, in the *wall* test, and for the three cameras, we have very small distance errors. Regarding the dispersion parameters, the small values of MSE and std indicate a concentrated distribution of the error distance values. For the *wall* test, and overall, D415 has the best performance.

In the test *empty*, the estimated distance to the plane is still approximately 0.63 m for the three cameras. The number of used points decreases for all cameras, though the D415 has the smallest decrease. The overall analysis of the results show that also in this case D415 has the best results.

Additionally, in Table 4, we have the *water full* test, where the estimated distances to planes are different from the previous tests. For the D415, we have a plane estimated at 0.56 m, the L515 estimates a plane at 0.67 m and the SR305 at 0.31 m. The number of points used to compute the estimates decreases for all cameras. The reduction is more significant for the L515 and SR305 cameras which use only 9% and 0.4% of points, respectively. Given the very small number, the data for the SR305 were not analyzed. Overall the results for D415 and L515 are similar.

Then, in the following table (Table 5) the results for the experiments with milk are presented. In some of the tests, the model parameters as well as the distances were not estimated given the size of the dataset. Overall, the estimates obtained with the D415 are the best in *water milk* tests. D515 is the only of the 3 cameras for which the number of points was considered sufficient to allow to estimate the plane parameters.

The following histograms (Figure 15, Figure 16 and Figure 17) represent the distribution of the number of points (relative frequency) with respect to the orthogonal distances to the estimated plane (relative orthogonal distance). It allows and enables a visual evaluation of the results presented in the tables. For example, and in the case of the D415 camera, the histograms show that the distribution of the points relative to the orthogonal distances is less spread out than those corresponding to the L515 and SR305 (for the same experiment).

### 5.3. Results and Discussion Regarding Plane Modeling Consistency

As aforementioned in Section 4.2.3, the consistency of the planes estimation was evaluated in terms of statistical parameters. More specifically, the standard deviations of the components of the normal to the plane and the spherical variance of the normal were estimated. Table 6 presents the results for the standard deviations and for the spherical variances for the cases of the three cameras and for the seven different experiments. Similarly to the the previous results (Section 5.2), the parameters were not estimated when the number of points was small.

One remark that can be made, from the analysis of the results, is related to the standard deviations of the components of the normal. The *z* component of the normal always has consistently a smaller standard deviation than the other two components. This results from the specific configuration of the setup, where all the planes are approximately perpendicular to the optical axes of the cameras.

The behavior of the standard deviations is similar for the cases of the three cameras (relative to the types of experiments) with the smallest values corresponding to the *wall* experiment. In all the other experiments the standard deviations increase, except for the case of *water milk4* (less transparent) where their orders of magnitude are similar to the values of the *wall* experiment.

Regarding the spherical variance, the smallest variance corresponds to the *wall* acquisition from D415 with an order of magnitude of O−6. All the other variances have a higher order of magnitudes. For the three cameras, the highest spherical variances corresponds to the acquisitions *water milk2* and *water milk3*.

The evaluation of the variation of the plane normals throughout the various experiments allows us to conclude that the D415 has smallest standard deviations and spherical variances in all the scenarios compared to L515 and SR305. Therefore, the D415 provides more stable estimates of the planes and so, more consistency/precision in depth estimations. As expected, the cameras more dependent on the laser/light reflection are less stable in scenarios with transparent and translucent surfaces.

### 5.4. Results and Discussion Regarding Depth Distribution

In Figure 18, the histograms that show the relative number of points as a function of the average estimated depths are presented.

It should be noted that the bars at 0m represent the relative frequency of the number of *invalid points*, that is, points where the cameras were unable estimate a depth value with their value being set to zero. In the *wall* test, the three cameras have a distribution of the depth estimates with peaks of the relative frequencies at 0.63 m (corresponding to the distance between the camera and the wall). The peaks continue to occur at 0.63 m for the the *empty* test.

The results show that the transparency of the empty aquarium barely affects the estimation of the wall for the three cameras. On the other hand, when the aquarium is filled with water, depth estimation is significantly affected, especially for the SR305 camera and also for the L515. In the case of the L515 camera, the offset of the depth estimation can be explained by the time-of-flight operating principle that the camera uses. This is because the IR rays emitted by the sensor are refracted and their velocities in water are greatly attenuated. The speed of light in water is 2.26 × 10${}^{8}$ m/s while in air it is 2.99 × 10${}^{8}$ m/s. Thus, the flight time will be longer and the estimated distance will be greater than the actual distance. The addition of milk to the water in the aquarium adds to the problems with the estimates obtained with the SR305 and L515 cameras. This behavior results from the phenomenon shown in Figure 8. That is, the infra red light used by the cameras undergo significant multiple scattering along the aquarium.

## 6. Conclusions

In this paper, we have presented an experimental assessment of depth estimation performance for three low-cost RGB-D cameras, with different operating principles, in scenes with transparent and translucent objects. After reviewing the literature, no research has been found to evaluate and compare these three state-of-the-art cameras, or similar cameras with these operating principles in scenarios with transparent and translucent objects. The authors [34] have also presented an evaluation comparing these same cameras but in a different context, where the cameras have to estimate depth of a white wall at different distances. The operating principles of each camera have been investigated to better understand the relative disadvantages or advantages of each camera for the specific experiments of the evaluation. We have proposed a set of different experiments that have been performed in a controlled environment, a darkroom with controlled light conditions. In the experiments, an aquarium has been used to introduce different levels of transparency and translucency, having been filled with water and with water–milk mixtures with four different concentrations. On the white wall behind the aquarium, a black cardboard has been used to create a difficult surface for the active estimation of depth. On the other hand it also enabled the visual perception of the degree of translucency of the mixture of water and milk. Depth image acquisitions were performed and converted into point clouds to facilitate the processing of depth data. The performance of the depth estimates has been evaluated in terms of failure rate, statistical analysis of point-to-planes orthogonal distances, and consistency of the estimated normal vectors. Furthermore, a qualitative analysis of the depth distribution is performed. The results obtained through the different analyses provide the following conclusions:-The black cardboard was a significant constraint for the SR305 and L515 cameras, being more significant for the SR305;-The transparency due to the aquarium glass walls do not affect the depth estimation in the three cameras. The wall is detected by all the cameras in a consistent way;-When the aquarium is filled with water, we have a more complex degree of transparency that negatively impacts depth estimation. The SR305 and the L515 are the most affected, having a high failure rate and very few *coherent points* to estimate consistent planes;-The water–milk mixture is a true issue even for the D415 camera. Still, the D415 continues to outperform the other two. The translucency of the water–milk mixture transmits light diffused in all directions, which drastically interferes with the entire operating principle of the SR305 and L515 cameras that emit an infra-red light that which ends up being interfered.

In general, comparing the precision of the three cameras, the D415 camera has advantage over the L515 and SR305 when facing transparent or translucent objects. As expected, the structured light technology of SR305 is drastically affected by transparency or translucency. The transparent/translucent objects inserted between the SR305 and the actual target work as barriers to the patterns preventing these from being projected in the target. Since L515 is highly dependent on the signal to noise ratio (SNR), a change in the quality of the returned signal, caused by the interference of transparent or translucent objects, significantly affects the depth estimates. On the other hand, the two cameras of D415 makes the system less sensitive to external factors (the transmittance of the material in this case), since the two patterns being compared will be affected by the same factors in equal proportion.

## Figures and Tables

**Figure 1 sensors-22-07378-f001:**
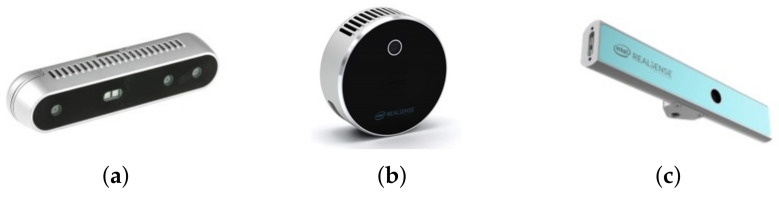
Intel^®^ RealSense™ cameras considered in this work. (**a**) Intel^®^ RealSense™ D415. (**b**) Intel^®^ RealSense™ L515. (**c**) Intel^®^ RealSense™ SR305.

**Figure 2 sensors-22-07378-f002:**
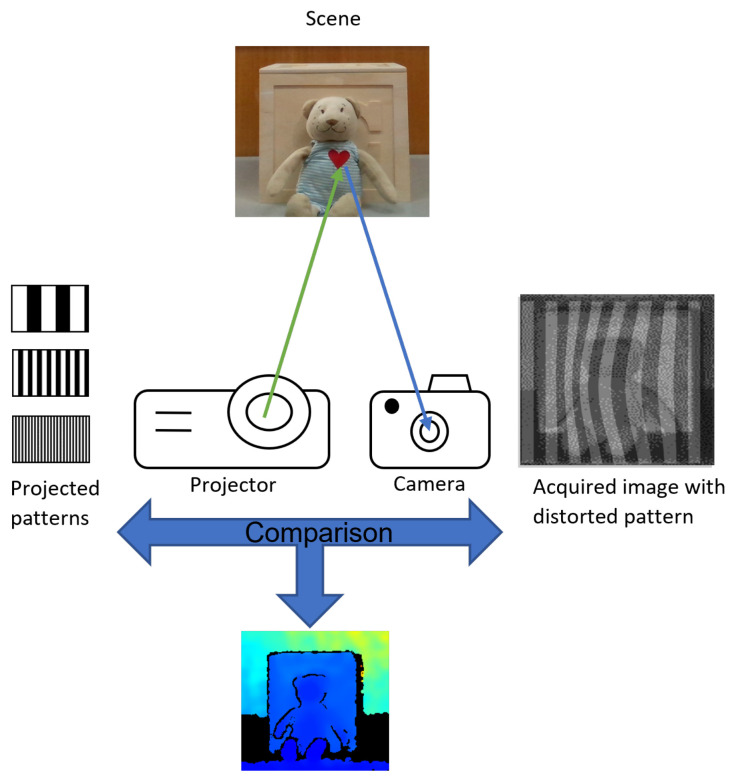
An illustration of a Structured Light sensor: the projector emits one or more patterns sequentially onto the scene. These patterns are warped by the scene, reflected back and captured by the camera. The scene depth is estimated by comparing the projected patterns and the distorted ones acquired by the camera.

**Figure 3 sensors-22-07378-f003:**
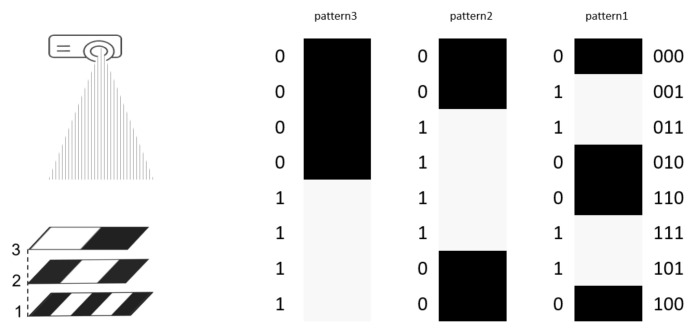
Gray Code patterns (N=3).

**Figure 4 sensors-22-07378-f004:**
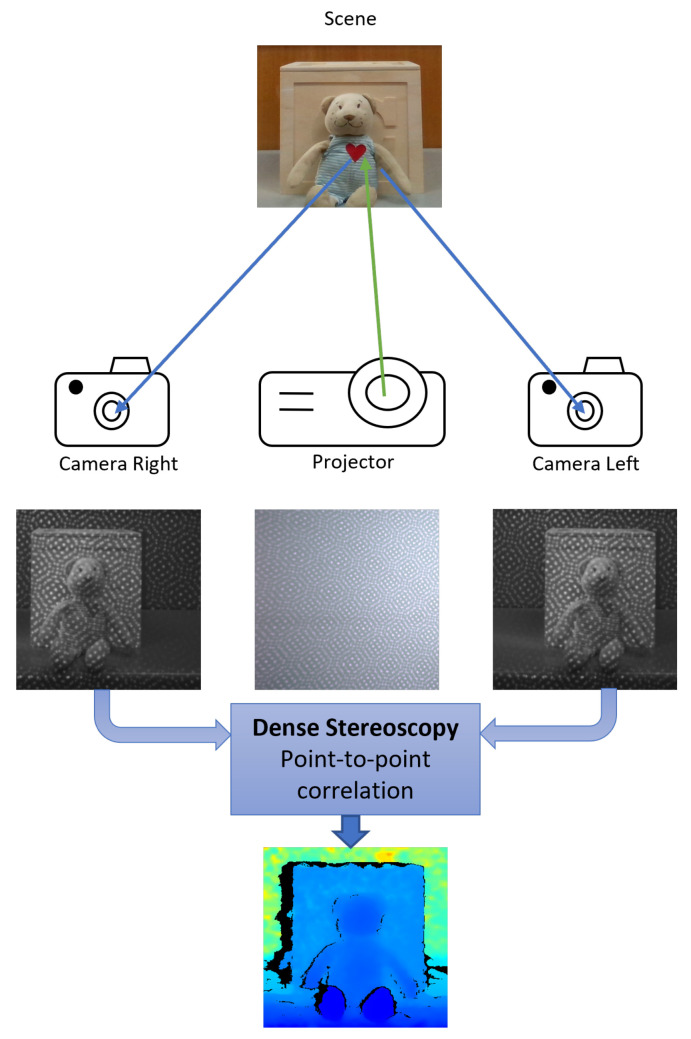
An illustration of an Active Stereoscopy sensor: The projector emits one random pattern onto the scene. This pattern adds additional features, thus, more correspondences. The scene depth is then estimated by triangulation using the disparity map.

**Figure 5 sensors-22-07378-f005:**
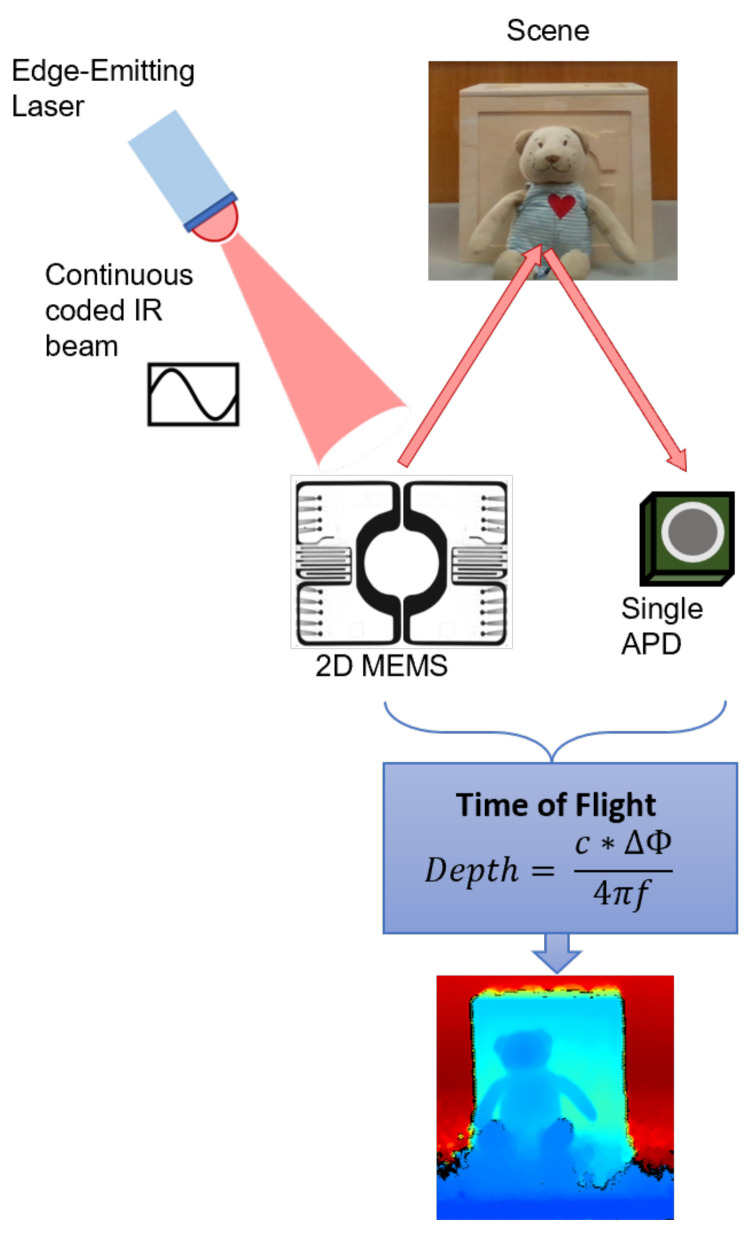
An illustration of the Time of Light technology present in L515 camera: an Edge-Emitting laser (EEL) emits a continuous modulated IR beam that is scanned across the entire FOV through a 2D MEMS mirror. The scene reflects the IR which is received by a single Avalanche Photodiode (APD). The scene depth is estimated by measuring the phase delay between the transmitted and received light beam.

**Figure 6 sensors-22-07378-f006:**
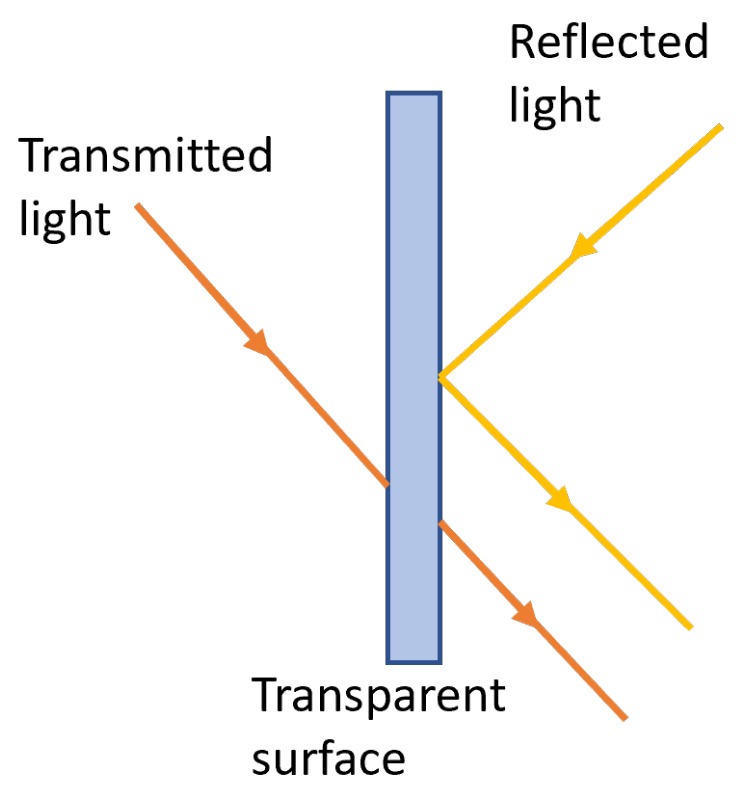
Light emission from a transparent surface is in general a combination of reflected and transmitted light.

**Figure 7 sensors-22-07378-f007:**
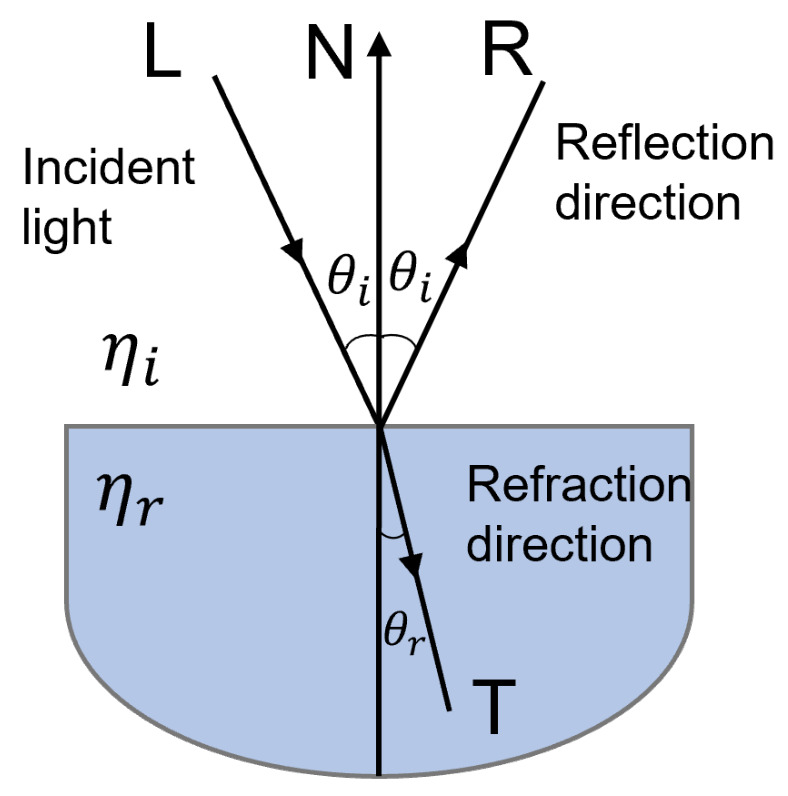
An incident light ray coming from the medium with refractive index $\eta_{i}$ is reflected and refracted through the directions R and T, respectively, upon a surface with refractive index $\eta_{r}$.

**Figure 8 sensors-22-07378-f008:**
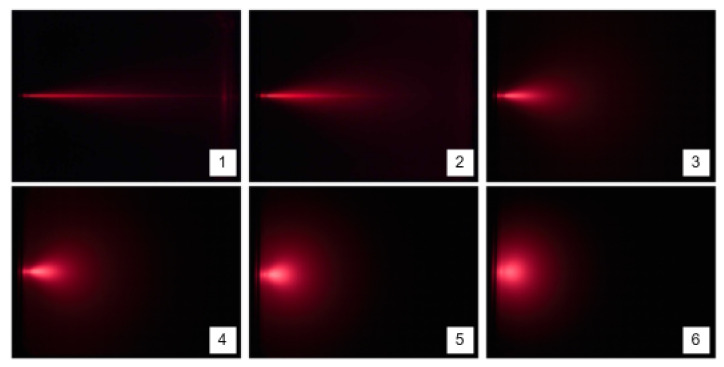
Effects of multiple scattering in colloids [56]. The first picture shows us an aquarium filled with water and a collimated laser beam running the full length of it. When milk is sequentially added to water (2–6), the solution becomes colloidal and there is spatial and angular spreading of the laser beam (Tyndall effect). At the point of picture 6, the laser spreading is essentially diffuse and has no longer the directionality present in the first picture.

**Figure 9 sensors-22-07378-f009:**
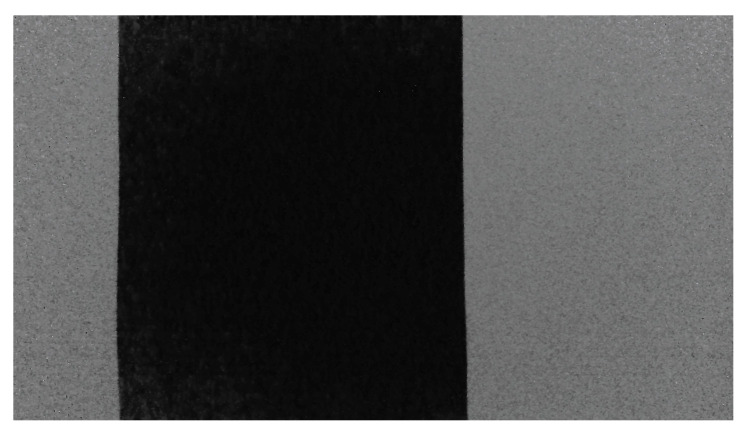
An image captured by the (RGB) SR305 camera where the black cardboard on the white wall (on the left) is visible.

**Figure 10 sensors-22-07378-f010:**
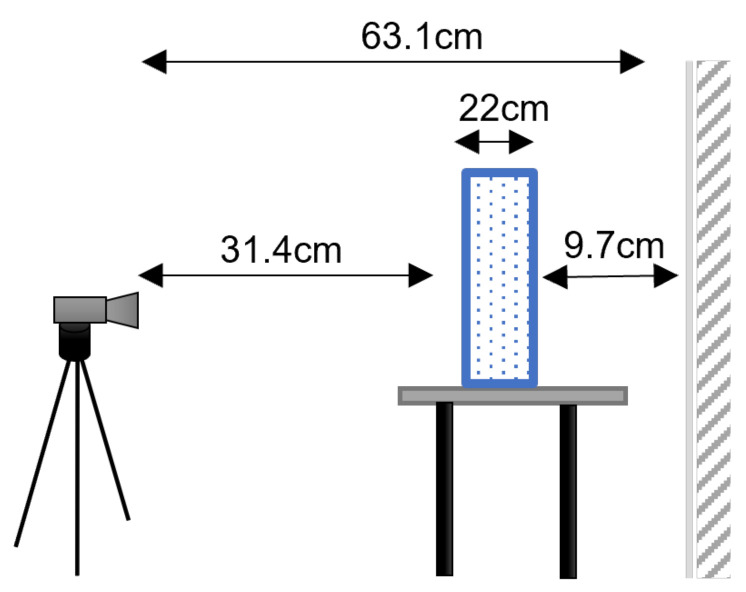
Illustration of the experimental setup. The aquarium is located 9.7 cm away from the wall and the camera is 31.4 cm away from the front surface of the aquarium. Since the width of the aquarium is 22 cm, the distance between the camera and the wall is 63.1 cm.

**Figure 11 sensors-22-07378-f011:**
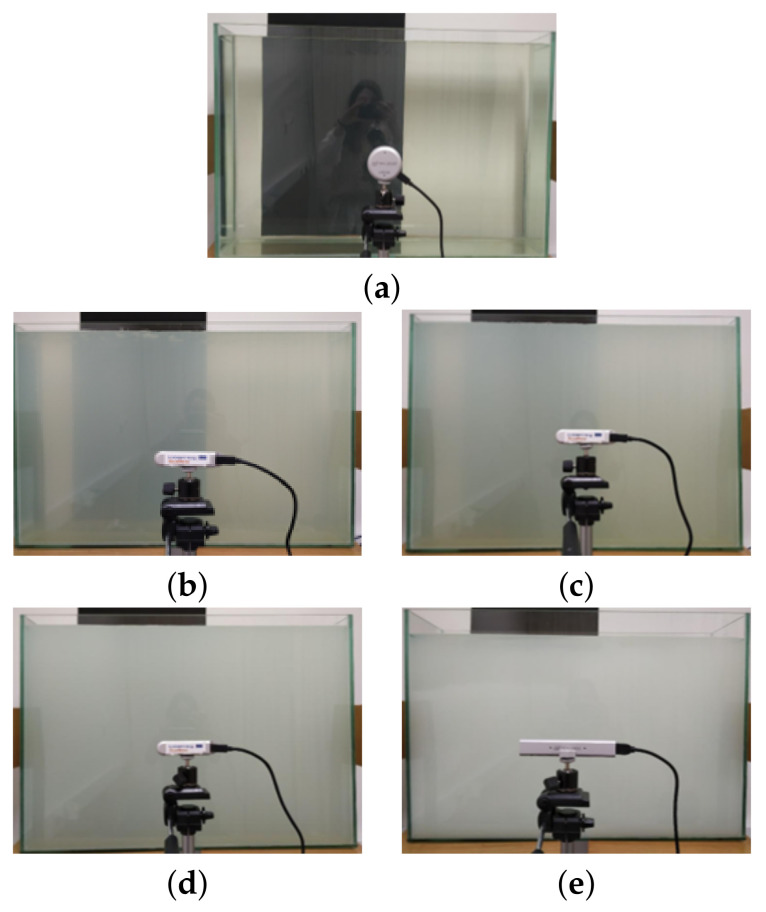
Pictures showing the aquarium at different stages of the experiment. (**a**) *water full*. (**b**) *water milk1*. (**c**) *water milk2*. (**d**) *water milk3*. (**e**) *water milk4*.

**Figure 12 sensors-22-07378-f012:**
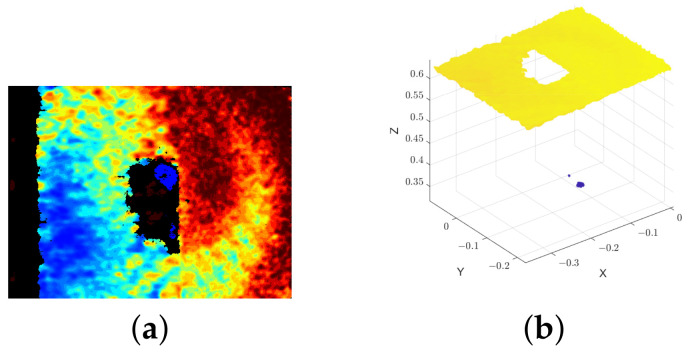
Example of a depth frame from the acquisition D415—*empty* converted to a point cloud. (**a**) Depth frame (by D415 camera) colorized in the function of depth values. (**b**) Point cloud.

**Figure 13 sensors-22-07378-f013:**
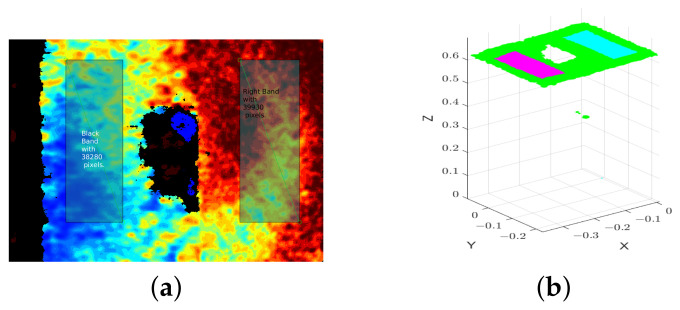
Example of a depth frame with segmented regions and respective point cloud with the segmented regions from the acquisition D415—*empty*. (**a**) Depth frame with segmented bands/regions. (**b**) Point cloud with segmented bands/regions. The *left band* is colored in magenta and the *right band* is colored in cyan.

**Figure 14 sensors-22-07378-f014:**
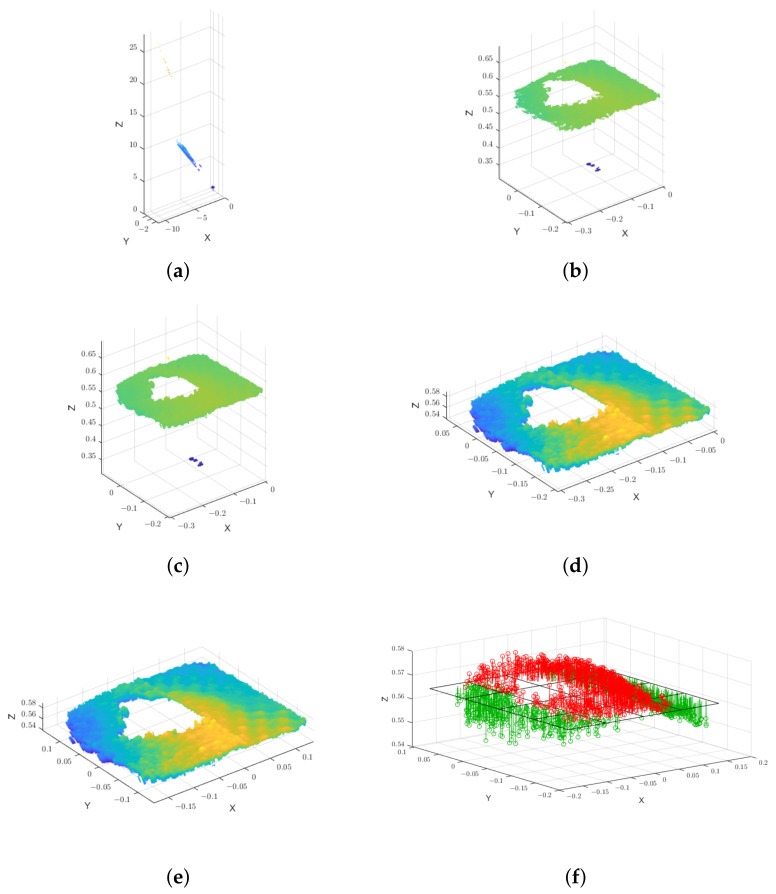
Methodology steps for the case of D415—*water full*. (**a**) ptCloud_raw. (**b**) ptCloud_filt. (**c**) ptCloud_concat. (**d**) ptCloud_in. (**e**) ptCloud_cent. (**f**) Estimated plane and orthogonal distances. Positive orthogonal distances are colored in red and negative in green. For visualization purposes, (**f**) contains a sub-sample of the estimated orthogonal distances. That is, the distances were displayed with a ratio of $\frac{1}{1000}$ regarding the total number of distances.

**Figure 15 sensors-22-07378-f015:**
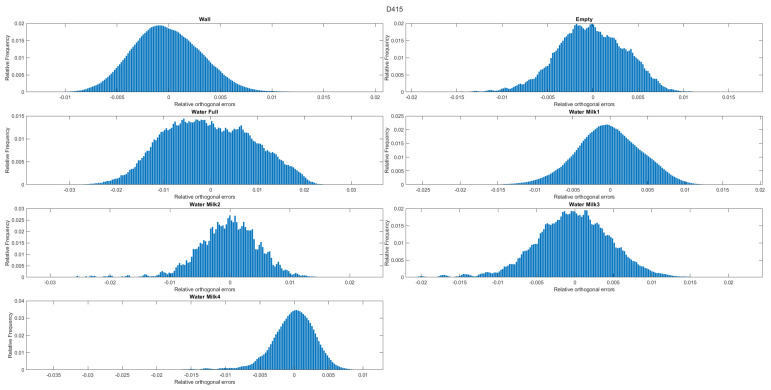
Histogram of orthogonal distances for the D415 camera.

**Figure 16 sensors-22-07378-f016:**
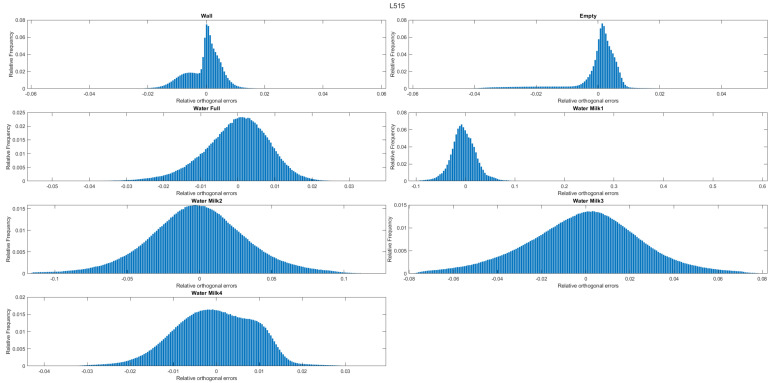
Histogram of orthogonal distances for the L515 camera.

**Figure 17 sensors-22-07378-f017:**
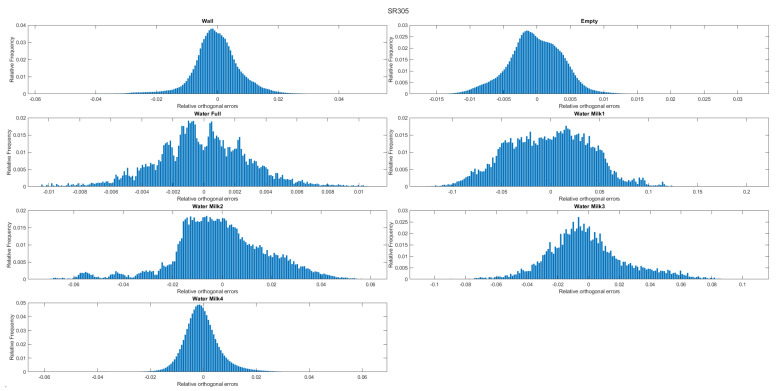
Histogram of orthogonal distances for the SR305 camera.

**Figure 18 sensors-22-07378-f018:**
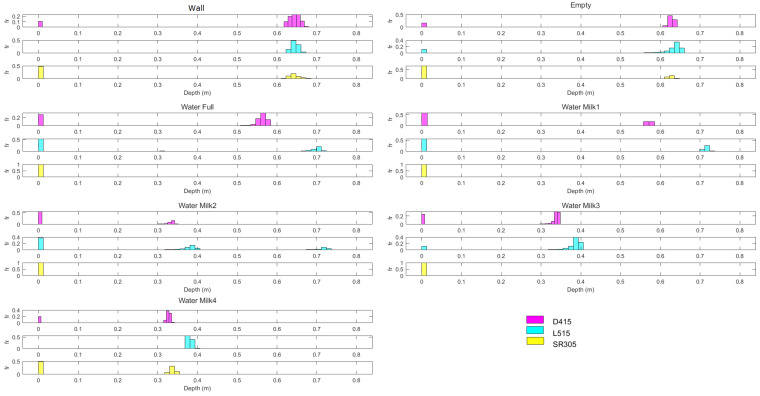
Histograms for each acquisition (allowing a comparison between cameras): each histogram represents the relative frequency for each average depth value estimated by each camera.

**Table 1 sensors-22-07378-t001:** Main specifications of the Intel^®^ RealSense™ cameras considered in this work.

	D415	L515	SR305	Units
Depth Measurement technology	Active Stereoscopy	Time-of-Flight	Structured Light	-
Image Sensor Technology	Rolling Shutter	MEMS mirror scan	Global Shutter	
IR max. resolution	1280 × 720	1024 × 768	640 × 480	(pix)
RGB max. resolution	1920 × 1080	1920 × 1080	1920 × 1080	(pix)
Maximum frame rate	90	30	60	(fps)
Baseline	55	-	-	(mm)
Depth Field of view (FOV)	H: 65 ± 2|V: 40 ± 1	H: 70 ± 2|V: 55 ± 2	H: 69 ± 3|V: 54 ± 2	(°)
Measurement range	0.3–10	0.25–9	0.2–1.5	(m)
Dimension	99 × 20 × 23	61 × 26	139 × 26.13 × 12	(mm)
Weight	72	95	70	(gr)

**Table 2 sensors-22-07378-t002:** The structure of the point clouds at different processing stages for the case of D415—water full.

	nº Points	X Limits	Y Limits	Z Limits
ptCloud_raw	307,200	[−11.2596, 0.0066]	[−2.5039, 0.6715]	[0, 27.7300]
ptCloud_filt	233,977	[−0.3198, 0.0066]	[−0.2086, 0.0763]	[0.3080, 0.7000]
ptCloud_concat	18,229,200	[−0.3068, 0.0067]	[−0.2090, 0.0770]	[0.3090, 0.7000]
ptCloud_in	18,126,707	[−0.3068, 0.0067]	[−0.2090, 0.0770]	[0.5390, 0.5890]
ptCloud_cent	18,126,707	[−0.1788, 0.1346]	[−0.1421, 0.1439]	[0.5390, 0.5890]

**Table 3 sensors-22-07378-t003:** Average percentage of *invalid points* for the *right band* and *left band*.

	$D415_{wall}$	$D415_{empty}$	$D415_{w\_full}$	$D415_{w\_milk1}$	$D415_{w\_milk2}$	$D415_{w\_milk3}$	$D415_{w\_milk4}$
${\bar{f\%}}_{rb}$	0.0041	0.0150	0.0177	0.0001	37.9954	0.6936	0.0000
${\bar{f\%}}_{lb}$	0.0000	0.0011	0.2806	81.4556	2.3833	1.0275	0.0147
	$L515_{wall}$	$L515_{empty}$	$L515_{w\_full}$	$L515_{w\_milk1}$	$L515_{w\_milk2}$	$L515_{w\_milk3}$	$L515_{w\_milk4}$
${\bar{f\%}}_{rb}$	0.0000	0.0000	0.0321	1.4239	87.9116	2.5124	0.0000
${\bar{f\%}}_{lb}$	0.0014	8.0577	99.8891	99.9018	0.7502	0.0417	0.0000
	$SR305_{wall}$	$SR305_{empty}$	$SR305_{w\_full}$	$SR305_{w\_milk1}$	$SR305_{w\_milk2}$	$SR305_{w\_milk3}$	$SR305_{w\_milk4}$
${\bar{f\%}}_{rb}$	0.0000	0.0021	100.0000	100.0000	100.0000	100.0000	0.0378
${\bar{f\%}}_{lb}$	100.0000	100.0000	100.0000	100.0000	100.0000	100.0000	0.0198

${\bar{f\%}}_{rb}$—Average of percentage of *invalid points* regarding the RoI *right band* (*rb*). ${\bar{f\%}}_{lb}$—Average of percentage of *invalid points* regarding the RoI *left band* (*lb*).

**Table 4 sensors-22-07378-t004:** Results for orthogonal distances to planes for the experiments *wall*, *empty* and *water full*.

Camera	Acquisition	Used Points ${}^{1}$	Dist. Plane ${}^{2}$	$\tilde{x}\left(\epsilon \right)$ ${}^{3}$	$\bar{x}\left(\epsilon \right)$ ${}^{4}$	$\sigma \left(\epsilon \right)$ ${}^{5}$	MSE(ϵ) ${}^{6}$	$\tilde{x}\left(\left|\epsilon \right|\right)$ ${}^{7}$	$\bar{x}\left(\left|\epsilon \right|\right)$ ${}^{8}$	$\sigma \left(\left|\epsilon \right|\right)$ ${}^{9}$
D415	wall	88.7129%	0.6393 m	$-9.34\times 10^{-5}$	$-2.50\times 10^{-19}$	0.0021	$4.50\times 10^{-6}$	0.0015	0.0017	0.0013
	empty	80.2240%	0.6266 m	$-2.01\times 10^{-5}$	$6.03\times 10^{-20}$	0.0024	$5.68\times 10^{-6}$	0.0016	0.0019	0.0014
	water full	59.3398%	0.5630 m	$-2.40\times 10^{-4}$	$-7.53\times 10^{-20}$	0.0053	$2.80\times 10^{-5}$	0.0040	0.0044	0.0030
L515	wall	97.7275%	0.6441 m	$6.55\times 10^{-4}$	1.70×10−20	0.0038	$1.45\times 10^{-5}$	0.0023	0.0029	0.0025
	empty	67.6309%	0.6414 m	0.0014	$1.01\times 10^{-19}$	0.0059	$3.51\times 10^{-5}$	0.0021	0.0036	0.0047
	water full	8.9310%	0.6748 m	$4.81\times 10^{-4}$	$3.96\times 10^{-20}$	0.0060	$3.55\times 10^{-5}$	0.0038	0.0046	0.0037
SR305	wall	48.9993%	0.6407 m	$4.13\times 10^{-5}$	$5.75\times 10^{-20}$	0.0055	$3.01\times 10^{-5}$	0.0027	0.0038	0.0040
	empty	28.2650%	0.6248 m	$-1.89\times 10^{-5}$	$7.50\times 10^{-19}$	0.0025	$6.21\times 10^{-6}$	0.0016	0.0020	0.0015
	water full	0.3529%	0.3075 m	-	-	-	-	-	-	-

^1^ Percentage of used points regarding the initial points (of the original point cloud). ^2^ Distance from plane to camera. ^3^ Median of the orthogonal distances *ϵ*. ^4^ Mean of the orthogonal distances *ϵ*. ^5^ Standard deviation of the orthogonal distances *ϵ*. ^6^ Mean Squared Error of the orthogonal error distances. ^7^ Median of the absolute orthogonal distances *ϵ*. ^8^ Mean of the absolute orthogonal distances *ϵ*. ^9^ Standard deviation of the absolute orthogonal distances *ϵ*.

**Table 5 sensors-22-07378-t005:** Results for orthogonal distances to planes for the experiments *water milk1*, *water milk2*, *water milk3* and *water milk4*.

Camera	Acquisition	Used Points ${}^{1}$	Dist. Plane ${}^{2}$	$\tilde{x}\left(\epsilon \right)$ ${}^{3}$	$\bar{x}\left(\epsilon \right)$ ${}^{4}$	$\sigma \left(\epsilon \right)$ ${}^{5}$	MSE(ϵ) ${}^{6}$	$\tilde{x}\left(\left|\epsilon \right|\right)$ ${}^{7}$	$\bar{x}\left(\left|\epsilon \right|\right)$ ${}^{8}$	$\sigma \left(\left|\epsilon \right|\right)$ ${}^{9}$
D415	water milk1	38.7015%	0.5681 m	$-2.51\times 10^{-5}$	$-1.39\times 10^{-19}$	0.0025	$6.03\times 10^{-6}$	0.0016	0.0020	0.0015
	water milk2	8.5417%	0.3411 m	$8.56\times 10^{-5}$	$-3.49\times 10^{-20}$	0.0018	$3.11\times 10^{-6}$	0.0011	0.0013	0.0011
	water milk3	36.8538%	0.3419 m	$2.49\times 10^{-5}$	$2.21\times 10^{-19}$	0.0017	$2.82\times 10^{-6}$	0.0011	0.0013	0.0010
	water milk4	78.2705%	0.3257 m	$1.05\times 10^{-4}$	$3.03\times 10^{-19}$	0.0011	$1.28\times 10^{-6}$	$6.50\times 10^{-4}$	$8.21\times 10^{-4}$	$7.81\times 10^{-4}$
L515	water milk1	1.1019%	0.4221 m	-	-	-	-	-	-	-
	water milk2	21.9079%	0.3867 m	$-5.01\times 10^{-5}$	$-4.17\times 10^{-19}$	0.0132	$1.75\times 10^{-4}$	0.0083	0.0103	0.0103
	water milk3	54.7113%	0.3921 m	$3.37\times 10^{-4}$	$-1.52\times 10^{-19}$	0.0101	$1.03\times 10^{-4}$	0.0065	0.0080	0.0063
	water milk4	99.3779%	0.3798 m	$6.29\times 10^{-5}$	$9.68\times 10^{-21}$	0.0035	$1.24\times 10^{-5}$	0.0025	0.0029	0.0021
SR305	water milk1	0.0918%	0.0177 m	-	-	-	-	-	-	-
	water milk2	0.1139%	0.0519 m	-	-	-	-	-	-	-
	water milk3	0.0579%	0.0222 m	-	-	-	-	-	-	-
	water milk4	45.4095%	0.3370 m	$-1.45\times 10^{-4}$	$-1.51\times 10^{-19}$	0.0025	$6.03\times 10^{-6}$	0.0013	0.0018	0.0017

^1^ Percentage of used points regarding the initial points (of the original point cloud). ^2^ Distance from plane to camera. ^3^ Median of the orthogonal distances e. 4 Mean of the orthogonal distances *ϵ*. ^4^ Mean of the orthogonal distances *ϵ*. ^5^ Standard deviation of the orthogonal distances *ϵ*. ^6^ Mean Squared Error of the orthogonal error distances. ^7^ Median of the absolute orthogonal distances *ϵ*. ^8^ Mean of the absolute orthogonal distances *ϵ*. ^9^ Standard deviation of the absolute orthogonal distances *ϵ*.

**Table 6 sensors-22-07378-t006:** Camera precision in terms of plane modelling consistency: Standard deviations of the normal components and spherical variance of the normal.

Camera	Acquisition	$\sigma_{n_{x}}$ ${}^{1}$	$\sigma_{n_{y}}$ ${}^{2}$	$\sigma_{n_{z}}$ ${}^{3}$	$V_{\mathrm{sph}}$ ${}^{4}$
D415	wall	$3.4\times 10^{-4}$	$2.4\times 10^{-4}$	$3.2\times 10^{-5}$	$7.2\times 10^{-6}$
	empty	$4.3\times 10^{-4}$	$4.3\times 10^{-4}$	$1.8\times 10^{-5}$	$1.1\times 10^{-4}$
	water full	$1.3\times 10^{-3}$	$1.1\times 10^{-3}$	$7.9\times 10^{-5}$	$3.1\times 10^{-4}$
	water milk1	$1.7\times 10^{-3}$	$1.1\times 10^{-3}$	$1.9\times 10^{-4}$	$1.2\times 10^{-4}$
	water milk2	$4.4\times 10^{-3}$	$2.4\times 10^{-3}$	$1.9\times 10^{-4}$	$3.2\times 10^{-3}$
	water milk3	$2.1\times 10^{-3}$	$1.9\times 10^{-3}$	$8.4\times 10^{-5}$	$2.2\times 10^{-3}$
	water milk4	$3.2\times 10^{-4}$	$2.6\times 10^{-4}$	$2.3\times 10^{-5}$	$1.3\times 10^{-5}$
L515	wall	$8.0\times 10^{-4}$	$5.4\times 10^{-4}$	$4.0\times 10^{-5}$	$1.3\times 10^{-4}$
	empty	$1.0\times 10^{-3}$	$4.4\times 10^{-4}$	$3.4\times 10^{-5}$	$3.1\times 10^{-4}$
	water full	$4.1\times 10^{-3}$	$2.8\times 10^{-3}$	$1.3\times 10^{-4}$	$8.8\times 10^{-3}$
	water milk1	-	-	-	-
	water milk2	$1.1\times 10^{-2}$	$8.1\times 10^{-3}$	$8.9\times 10^{-4}$	$1.0\times 10^{-2}$
	water milk3	$3.6\times 10^{-3}$	$3.4\times 10^{-3}$	$5.0\times 10^{-5}$	$6.7\times 10^{-2}$
	water milk4	$1.1\times 10^{-3}$	$7.3\times 10^{-4}$	$2.5\times 10^{-5}$	$1.4\times 10^{-3}$
SR305	wall	$9.7\times 10^{-3}$	$5.6\times 10^{-4}$	$4.3\times 10^{-4}$	$1.3\times 10^{-2}$
	empty	$9.2\times 10^{-3}$	$3.6\times 10^{-4}$	$3.2\times 10^{-4}$	$1.4\times 10^{-2}$
	water full	$2.0\times 10^{-2}$	$1.2\times 10^{-2}$	$4.1\times 10^{-4}$	$3.3\times 10^{-1}$
	water milk1	-	-	-	-
	water milk2	-	-	-	-
	water milk3	-	-	-	-
	water milk4	$4.7\times 10^{-3}$	$6.3\times 10^{-4}$	$2.7\times 10^{-4}$	$1.5\times 10^{-3}$

^1^ Standard deviation of the x component of the normal given by the estimated plane model. ^2^ Standard deviation of the y component of the normal given by the estimated plane model. ^3^ Standard deviation of the z component of the normal given by the estimated plane model. ^4^ Spherical variance of the normal given by the estimated plane model.

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
