# Peer review of "An Experimental Assessment of Depth Estimation in Transparent and Translucent Scenes for Intel RealSense D415, SR305 and L515"

_sensors, 2022, doi:10.3390/s22197378_

Round 1

Reviewer 1 Report

My comments are:

(1) This is a good job for readers in the field;

(2) The authors can add: from the analysis of structural light imaging principle, the imaging differences of different devices in different media environments;

(3) The reference format needs to be further improved

Reviewer 2 Report

This paper analyses the performance of three different RealSense™ RGB-D cameras regarding depth estimation in challenging scenarios comprising transparency, semi- transparency and refraction/scattering.

1)    The title is “… Intel RealSense D415, SR305 and L515”. However in the abstract D415, D435 and L515 were mentioned (lines 3 and 4). So what are “three different RealSense™ RGB-D cameras” (line 7) ??

2)    In Figure 6, why the reflected light's incident light does not pass through the transparent object, is it any different from the incident light on the left? It is necessary to redraw this figure to make it easier to understand the statement "Light emission from a transparent surface is in general a combination of reflected and transmitted light".

3)    The author should add the order of each image in Figure 8.

4)    Please check the notation V/V% or %V/V (lines 406 -408)

5)    The author has divided the depth frame into two areas corresponding to black and white backgrounds as shown in Figure 13a. The question is why are the dimensions of the two regions different? In addition, the dimensions of these regions (left is large and right is smaller) are not similar to those of the two regions in the color image against the black and white background, respectively (Figure 11).

6)    The paper also shows two segmented RoIs. However, it does not explain the rationale for choosing the dimensions of the two ROI regions (line 430). Why choose the dimensions of these two regions like that, or why not choose the dimensions such as those of the two regions in the color image? When comparing the dimensions of the ROI regions corresponding to three different types of Cameras (In Appendix A), it also shows that their dimensions are not the same. So does it have any effect on the results in Tables 3,4,5 and 6.

Reviewer 3 Report

This paper designs an experimental framework and evaluates the depth estimation performance of three cameras in transparent and translucent scenes. The description of the paper is rigorous, the level is clear, the data is detailed, and the experimental results are credible. The author needs to check and explain the following issues to improve the quality of the paper.

(1)There are some format problems in the paper:

The format and position of some graphs and tables are inconsistent with others. Equation 5 and equation 6 should be divided into two lines.

Line 593 does not end with a period.

(2) The original text of line 567 on page 19 reads "less 9% (89%)". Should we change 9% to 90%?

(3) This paper expounds the camera imaging principle, refraction principle and experimental effect respectively, but it does not analyze and deduce how the camera principle is affected by the transparent medium when explaining the experimental effect, which is not convincing.

It is suggested to review after modification.

Round 2

Reviewer 2 Report

This paper analyses the performance of three different RealSense™ RGB-D cameras regarding depth estimation in challenging scenarios comprising transparency, semi- transparency and refraction/scattering.

Please, a new contribution of the paper should be described in the abstract and the introduction.

Author Response

Thank you for your comment.

In the abstract (line 10) was added: " The main new contribution of this paper is a full evaluation and comparison between the three Intel RealSense cameras in scenarios with transparency and translucency."  

In addition, we also added (line 15): " The evaluation, based on repeatability/precision and statistical distribution of the acquired depth, allows us to compare the three cameras and conclude that Intel RealSense D415 has overall the best behavior namely in what concerns the statistical variability (also known as precision or repeatability) and also in what concerns valid measurements. "

Regarding the Introduction,  there is already a detailed list of the new contributions of the paper.

Reviewer 3 Report

Thank the author for answering my questions, but the manuscript was not modified enough. In the last paragraph, the connection between the result and the principle is slightly simple, and it is suggested to enrich the inference process.

It is suggested to receive this paper after the above modifications.

Author Response

In this paper three Intel RealSense cameras, based on different depth measurement principles, were compared and evaluated for scenarios with transparency and/or translucency. While the measurement principles are different, it is not possible to draw any direct conclusions relative to the physical principles, since the physical and optical parameters of the cameras are different and also data is obtained after significant internal processing whose details are not fully known. Therefore only conclusions relative to the cameras themselves can be drawn. Overall, the camera with the best repeatability/statistical variability is the D415. The D415 is also the camera with smaller invalid measurements for transparency/translucency scenarios.

In the Results section (lines 663-668), we added the following: " In the case of the L515 camera, the offset of the depth estimation can be explained by the time-of-flight operating principle that the camera uses. This is because the IR rays emitted by the sensor are refracted and their velocities in water are greatly attenuated. The speed of light in water is 2.26x108m/s while in the air it is 2.99x108m/s. Thus, the flight time will be longer and the estimated distance will be greater than the actual distance. "